# Resulting Effect of the p-Type of ZnTe: Cu Thin Films of the Intermediate Layer in Heterojunction Solar Cells: Structural, Optical, and Electrical Characteristics

**DOI:** 10.3390/ma16083082

**Published:** 2023-04-13

**Authors:** Moustafa Ahmed, Ahmed Alshahrie, Essam R. Shaaban

**Affiliations:** 1Department of Physics, Faculty of Science, King Abdulaziz University, P.O. Box 80203, Jeddah 21589, Saudi Arabia; 2Center of Nanotechnology, King Abdulaziz University, Jeddah 21589, Saudi Arabia; 3Physics Department, Faculty of Science, Al-Azhar University, Assiut P.O. Box 71452, Egypt

**Keywords:** ZnTe:Cu thin films, microstructural parameters, optical properties, electrical characteristics, heterojunction solar cells

## Abstract

The microstructural, electrical, and optical properties of Cu-doped and undoped ZnTe thin films grown on glass substrates are covered in this article. To determine the chemical makeup of these materials, both energy-dispersive X-ray (EDAX) spectroscopy and X-ray photoelectron spectroscopy were employed. The cubic zinc-blende crystal structure of ZnTe and Cu-doped ZnTe films was discovered using X-ray diffraction crystallography. According to these microstructural studies, the average crystallite size increased as the amount of Cu doping increased, whereas the microstrain decreased as the crystallinity increased; hence, defects were minimized. The Swanepoel method was used to compute the refractive index, and it was found that the refractive index rises as the Cu doping levels rises. The optical band gap energy was observed to decrease from 2.225 eV to 1.941 eV as the Cu content rose from 0% to 8%, and then slightly increase to 1.965 eV at a Cu concentration of 10%. The Burstein–Moss effect may be connected to this observation. The larger grain size, which lessens the dispersion of the grain boundary, was thought to be the cause of the observed increase in the dc electrical conductivity with an increase in Cu doping. In structured undoped and Cu-doped ZnTe films, there were two carrier transport conduction mechanisms that could be seen. According to the Hall Effect measurements, all the grown films exhibited a p-type conduction behavior. In addition, the findings demonstrated that as the Cu doping level rises, the carrier concentration and the Hall mobility similarly rise, reaching an ideal Cu concentration of 8 at.%, which is due to the fact that the grain size decreases grain boundary scattering. Furthermore, we examined the impact of the ZnTe and ZnTe:Cu (at Cu 8 at.%) layers on the efficiency of the CdS/CdTe solar cells.

## 1. Introduction

Silicon-based photovoltaic solar cells have dominated the market; however, their cost is high due to the manufacturing process. Therefore, the way forward is to develop thin- film solar cells using low-cost attractive materials from II–VI compounds that are grown by cheaper, scalable and manufacturable techniques. II–VI compounds are widely used in solid-state devices such as photovoltaic cells, optoelectronic devices, photoconductive and photo-electric devices [1,2,3]. ZnTe has, in particular, been studied for solar cell applications because of its optimum energy gap of 2.24 eV at room temperature and its low electron affinity of 3.53 eV. For example, ZnTe can be used in solar cells as a back surface field (BSF) layer and as a p-type semiconductor material for a CdTe/ZnTe structure [4,5]. VI-II composites are more important semiconductors for photoconductive and photoelectric device applications than lead halide-based perovskite semiconductors. This is because lead halide-based perovskite semiconductors possess non-stabilized efficiency records for thin-film solar cells, despite the fact that they are solution-processed semiconductors and do not even possess an ideal cell gap for single-junction solar cells [6,7,8].

Due to the minimal valence band offset of 0.05 eV between ZnTe and CdTe, a small layer of ZnTe is used as a reliable back contact in CdTe-based solar cells [9,10]. Among II-VI semiconductors, the ZnTe semiconductor is obviously a low carrier p-type system with an absorbance coefficient of 10^5^ cm^−1^ and a band gap energy of 2.24 eV [3]. Thin films of ZnTe are suitable materials for optoelectronic and electro-optic requests in the visible spectrum area because they can absorb visible light without the help of phonons [11,12]. ZnTe performs flawlessly as a useful intermediate layer in heterojunction solar cell technology when combined with a CdTe absorber and a metallic back connection. ZnTe semiconductors have been shown experimentally to be a low-resistive, dependable and effective back connection for polycrystalline ZnTe/CdTe/CdS/ITO solar cells [13,14]. Thin films of p-type CdTe have previously been reported to exhibit electron affinities and work functions of 4.28 eV and 5.78 eV, respectively. Combining metals and semiconductors makes it simple to produce a Schottky barrier. However, the main impediment to the manufacture of CdTe-based solar cells is a highly desirable ohmic contact. The demanding tasks required of back contacts, such as those indicated above, cannot be performed by metals. Hence, ZnTe is the ideal material to meet these needs and functions with tandem solar cells based on CdTe [6]. The high resistivity of the ZnTe system presents another bottleneck, which forces the adoption of a strongly doped p-type element in order to increase conductivity [15,16]. A promising p-type acceptor for this function is copper (Cu). The p-type nature endures even after Cu doping thanks to the ZnTe semiconductor’s self-compensating properties. According to previous studies, Cu doping of ZnTe results in the formation of an ohmic contact between the ZnTe and CdTe, making it a promising option for solar cell applications. Copper doping probably increases the number of hole carriers in the ZnTe semiconductor, which facilitates the formation of ohmic connections. In comparison to the Cu^2+^ state, p-type charge carriers are more likely to form when Cu species replace the Zn^2+^ species as the Cu^+^ state.

In the literature, there are a limited number of papers on Cu-doped ZnTe thin films and even fewer reports on the oxidation state of Cu dopants. As a result of our group’s continued search for novel materials, we demonstrate the fabrication of a thin Cu-doped ZnTe film sample. We provide the enhanced ZnTe:Cu thin-film synthesis parameters. We contrast the characteristics of the Cu-doped and undoped ZnTe thin films in terms of their structural, morphological and elemental composition analyses, and their optical, electronic and electrical characteristics. ZnTe thin films with Cu doping may be a good semiconducting material for the intermediate buffer layer in heterojunction solar cell technology because the introduction of shallow acceptor states lowers the bandgap energy while boosting the hole carrier concentration. Examining the impact of diluted Cu on the structural, optical and electrical characteristics of ZnTe films is the primary goal of this research. An evaluation of the performance of ZnTe-based heterojunctions with a CdS buffer layer and ZnTe:Cu as a back surface field (BSF) layer is the second goal. The third goal is to explain the manner in which the optical and electrical parameters change as well as how well the CdS/CdTe solar cells work.

## 2. Experimental Procedures

Using a traditional solid-state reaction method, various (ZnTe)_1−x_(Cu_2_Te)_x_ compositions (with x = 0, 0.02, 0.04, 0.06, 0.08 and 0.10) were fabricated. Aldrich laboratory-grade, high-purity (99.999%) ZnTe and Cu_2_Te powders were combined using a ball mill. The result was the following reaction:xCu_2_Te + (1 − x) ZnTe → Zn_1−x_Cu_x_Te 

The powders were combined to form a pellet in the form of a disc. The thin films were produced using these pellets as the raw materials; the same technique was recommended in our previous investigations of different compounds [16,17,18,19,20]. Thermal evaporation of the powdered samples onto the glass substrates at a pressure of roughly 10^−6^ Pa was used to produce the Zn_1-x_Cu_x_Te thin films. The films’ thickness was kept constant at 110 nm (±5 nm). The substrates were maintained at 400 K throughout the deposition procedure and the deposition rate was limited to 2 nm/s. A film produced at such a slow rate of deposition has the same composition as the starting bulk material [21,22]. A quartz crystal monitor DTM 100 connected to the vacuum system was used to regulate the film thickness and the evaporation rates. EDAX spectroscopy was used for the compositional investigations. It was possible to determine the composition of the suggested products with a relative error of less than 2%. ZnTe films with varied Cu contents were coded as (0C/ZT, 2C/ZT, 4C/ZT, 6C/ZT, 8C/ZT and 10C/ZT) for the undoped ZnTe and Cu-doped ZnTe samples. Figure 1a,b displays the EDAX spectra of the thin films formed from (a) pure ZnTe and (b) Zn_0.98_Cu_0.02_Te along with their weights and atomic percentages. Although the Cu’s signal was weak, it is still plainly detectable in the Zn_0.98_Cu_0.02_Te film. The outcomes of the EDXS analyses of all the films are also listed in Table 1. Using photoelectron spectroscopy (XPS), the characteristics of the band structure were investigated. By using X-ray diffraction diffractometry (Philips 1710) with a 2θ ranging between 5 and 70, the phase purity and crystal structure of the films were analyzed. The optical spectra of the formed ZnTe:Cu films were examined for transmission and reflection using a UV-Vis-NIR JASCO-670 two-beam spectrophotometer (Hachioji, Tokyo, Japan). Using the conventional four-point probe method, the electrical characteristics of the ZnTe:Cu films with various Cu concentrations were assessed. The solar simulator 1.5 worldwide spectrum (AM1.5G) and the Keithley 2400 supply meter system were used to assess the current density–voltage (J–V) properties of the solar cell device under normal test conditions.

## 3. Results and Discussion

### 3.1. Structural Analysis

The Rietveld refinement technique can be used to characterize crystalline materials [23]. The XRD pattern created by the X-ray diffraction of the powder sample contains reflections (intensity peaks) with specific diffraction angles. The Rietveld method was used to modify a theoretical line profile until it resembled the measured profile. Figure 2 depicts the Rietveld refinement for the ZnTe powder sample. The XRD patterns of the undoped and doped ZnTe thin films with various Cu concentrations are shown in Figure 3a. The films are polycrystalline deposits with lattice parameters of a = b = c = 6.103, which are part of the zinc-blende cubic structure (JCPDS No. 15-0746), as indicated by the peaks in the XRD patterns in Figure 3a.

Due to the narrow difference between Zn’s atomic radius of 75 pm and Cu’s atomic radius of 73 pm, it was expected that Cu could easily penetrate ZnTe to take Zn’s place in the crystal lattice, resulting in the formation of a single phase. This was corroborated by the fact that the doping of Cu into ZnTe films at various Cu concentrations had little to no impact on the crystal structure of ZnTe, which was evidenced by the absence of any secondary phase [20,24,25]. In the sample of 10C/ZT, a very small peak at 2θ = 27.51° belonged to traces of Te (JCPDS No. 001-0714), indicating an excess of Te at an atomic concentration of Cu = 10%. Additionally, copper diffusion via the copper telluride layer could account for the emergence of new peaks that correspond to those of metallic tellurium. Due to an excess or dopant limit for ZnTe, the tellurium was left alone. It is noteworthy that the XRD intensity-diffraction peaks increased with an increase in the Cu concentration in the film, due to the enhancement of the film’s crystallinity. The findings are in fair agreement with those in Reference [26]. Figure 3b displays the XRD peaks with the (111) index magnified, which moves to a higher diffraction angle as the Cu increases. This creates a residual stress by the copper interstitial replacement and the systematic incorporation of the Cu ions into the Zn lattice site, without changing its crystalline cubic structure.

The Miller indices (*hkl*) of the plane, interplanar spacing (*d_hkl_*), and the typical cubic structure lattice parameter “*a*” have the following relationship [27,28]:(1)$$d_{hkl}=\frac{a}{\sqrt{h^{2}+k^{2}+l^{2}}}$$

According to Bragg’s law, we have
(2)$$\lambda =2d_{hkl}\sin\theta$$
where *θ* is Bragg’s diffraction angle. The estimated lattice parameters *a* = *b* = *c* decrease with an increase in the Cu concentration and are in good agreement with the provided JCPDS data. The cause of the tiny range reduction in the lattice parameters may be due to the rise in stress brought on by differences in the ionic radii of Cu and Zn, which leads to lattice deformation.

The crystallite size D of the Zn_1−x_Cu_x_Te films was calculated using the Debye–Scherer’s equation [29,30]:(3)$$D_{v}=\frac{k\lambda }{\beta \cos(\theta )}\;$$

The Cu ion buildup altered the crystallite size expansion in the Zn_1−x_Cu_x_Te films, which could have sped up the nucleation and growth levels of the Cu-doped ZnTe films. The estimated size values are listed in Table 2. The lattice strain (ε) was determined using the Stoke and Wilson Equation [31,32]: (4)$$\varepsilon =\frac{\beta }{4\tan(\theta )}\;$$

*β* can be corrected from the following relationship
(5)$$\beta =\sqrt{\beta_{obs}^{2}-\beta_{std}^{2}}$$
where *β_obs_* is the film’s integral peak width and *β_std_* is the standard peak width (silicon). The lattice strain and average crystallite size of the Zn1-xCuxTe films (x = 0, 0.02, 0.04, 0.06, 0.08, and 0.10) are shown in Figure 4. The size of the crystallite grew with an increase in the Cu inclusion, while the lattice strain decreased. The trends in *D_v_* and *ε* may both be attributed to a change in the lattice size with the substitution of Cu by Zn, because the ionic radii of Zn and Cu are quite comparable.

### 3.2. X-ray Photoelectron Spectroscopy

Figure 5 shows the XPS scans of undoped (a) ZnTe and (b) ZnTe:Cu thin films. These samples contain (Zn), (Te) and (O), and the orbital presence of these essential elements are noted. The oxygen signal that was detected during the survey scan may have been caused by surface contamination from atmospheric oxygen. The high-outcome XPS measurements of each composition were further investigated separately.

Figure 5c,d shows the XPS of the O-1s peaks of the undoped and Cu-doped ZnTe film. The as-deposited ZnTe thin films’ deconvoluted O1s peaks (in terms of Gaussian fit) were examined at 532.5 eV and 535.4 eV. The O-C species were suggested by the peak at 535.4 eV. Following Cu doping in the ZnTe thin films, the deconvoluted peak locations for O1s were detected at 531.8 eV, 535.2 eV and 538.2 eV [33]. The fluctuation in the oxygen vacancy concentration in the deficient region caused the peak at 531.8 eV. Figure 5e,f shows the deconvoluted peaks for the Zn-2p1/2 and Zn-2p3/2, and for the ZnTe and ZnTe:Cu films, respectively. For the Zn2p spectra, a high resolution XPS was observed. Although they have lower binding energies (BE) than Zn_2_p_1/2_, Zn_2_p_3/2_ peaks frequently have higher intensities [34].

The fitted Zn_2_p_3/2_ peaks of the undoped ZnTe thin films were measured at 1023.25 (owing to Zn^2+^ oxidation) and 1022.63 eV, respectively; whereas 2p^1/2^ was found at peak central positions of 1049.80 eV (owing to Zn^2+^ oxidation) and 1047.4 eV (Zn metallic). After Cu was doped in the ZnTe matrix, the high-resolution XPS Zn_2_p peaks were pushed to higher-order values, mostly as a result of the doping effects [26].

After fitting, it was feasible to separate the Zn_2_p_1/2_ peak at 1053.3 eV and the higher intensity deconvoluted Zn_2_p_3/2_ peak at 1031.5 eV. The unbounded Zn species is associated with a computed Zn_2_p_3/2_ peak for the ZnTe:Cu films at eV 1023.7 and a Zn_2_p_1/2_ peak at eV 1046.8. The results indicate that a metallic zinc less than Zn^2+^ was observed with Cu doping. The Cu-doped ZnTe thin film enables the introduction of more Zn^2+^ states than the samples of undoped ZnTe thin film.

Figure 5 displays the high-resolution XPS spectra for the samples with and without Cu doping after a Te-3d devolution (g, h). Peaks at 587.7 eV, 575.1 eV and 571.5 eV were visible in the ZnTe thin films that were not doped. However, after Cu doping, peaks were discovered at 592.42 e, 588.1 eV, 585.6 eV, 579.11 eV, 575.4 eV, 573.1 eV and 571.10 eV. Significant Te_3_d_5/2_ and Te3d_3/2_ peaks were seen at locations of 586.7 and 575.1 for the un-doped, and 585.6 and 576.4 for those doped with copper; they were mostly brought on by oxide tellurium (TeO_2_, TeO_3_). For both samples, Te3d_3/2_ was discovered to be at a peak position of 571.1 eV. However, Te_3_d_5/2_ was found to be at a peak position of 582.22 eV only in the Cu-doped ZnTe thin film. The majority of the Te3d spectra reported in the literature have four peaks, whereas our undoped and doped samples showed different Te-3d spectra. The lack of a Te-3d_5/2_ XPS signal in the undoped ZnTe thin film may be a sign that fewer Zn species were bound to Te in the absence of Cu doping. Therefore, Cu is crucial in this situation for increasing the number of connections between the Te and Zn species. Owing to the integration of Cu into the matrix of the ZnTe films, the Auger peaks were visible. The ZnTe:Cu film with a high resolution of Cu2p had binding energies at 931 eV and 951 eV, in accordance with the Cu+ and Cu2+ oxidation states, respectively [26]. Another noticeable peak was found at 569.87 eV, which matches the Cu+ species described in [26,34]. Similarly, we discovered a Zn LMM Auger peak [26] at location 579.60 eV in the ZnTe:Cu films.

### 3.3. Optical Properties

Figure 6 plots the measured spectra of transmittance T(λ) and reflectance R(λ) (300–2500 nm) of the crystalline Cu-doped ZnTe films at various Cu doping levels. The usual near-infrared transmittance ranged from 75% to 87%. Consequently, the very transparent Cu-doped ZnTe films are used in n-type layer window applications for solar cells. That is, the XRD analysis clearly showed that an increasing concentration of Cu in the ZnTe host lattice indicates an improvement in crystallinity, which causes a considerable decline in transmittance. Swanepoel’s envelope method was utilized to determine the refractive index, n, of the thin films under investigation using the experimentally obtained transmission data [35,36]. For the optical transmittance, T(λ), of the ZnTe thin films, the application of the envelope technique utilizing fit exponential growth is presented in Figure 6. The same method was applied to the ZnTe:Cu thin films with different chemical compositions. The Swanepoel method’s index of refraction, n, can be determined at any wavelength using the following formula [37,38]:(6)n=[N+(N−2s)21/2]1/2
where
(7)N=2sTM−TmTMTm+s2+12, s=1Ts+(1Ts2−1)1/2
and where *T_M_* (*λ*) and *T_m_* (λ) are the transmittance maximum and minimum, respectively, and s is the substrate’s index of refraction. The envelopes of T_M_ and T_m_ are shown in Figure 7. The correlation between the wavelength and the refractive index, *n*, of the ZnTe:Cu thin films is shown in Figure 8. For the composites of ZnTe:Cu that exhibited normal dispersion behaviors, a decrease in *n* occurred with an increase in the wavelength. Furthermore, when the level of Cu content rose, the refractive index *n* typically increased. An increase in the number of atoms at interstitial locations could account for this by creating impurity scattering centers in the films under investigation [38,39]. Through the use of a number of empirical models, it was observed that the bandgap energy and the refractive index are inversely related; a decrease in the bandgap energy with an increase in the Cu content is associated with an increase in the refractive index [40].

The transmission curve also dropped at the absorption edge due to the transit of electrons from the valence band (VB) to the conduction band (CB). An additional investigation demonstrated that the basic energy gap narrows with an increase in Cu doping. As the Cu concentration of the ZnTe increases, the absorption edge redshifts. According to observations of transmission and reflection in the high absorption region, the absorption coefficient of a nanostructured Cu-doped ZnTe film with varied Cu dopants is derived from Figure 9 as an occupation of photon energy using the following relation [41]:(8)$$\alpha =\frac{1}{d}\ln\left[\frac{{\left(1-R\right)}^{2}+\sqrt{{\left(1-R\right)}^{4}+4{\left(TR\right)}^{2}}}{2T}\right]$$
where *d* is the film’s thickness. The films are suited for use as an absorber layer in solar cells due to their high absorption (10^5^ cm^−1^) properties.

The results show that the absorption increases with an increase in the Cu content. It can also be observed that when the degree of Cu incorporation increases, the absorption value drops at the absorption edge and shifts to a lower energy. Using Tauc’s model [42], the optical band gap energy of the Cu-doped films is determined from the following equation:(9)$${\left(\alpha h\nu \right)}^{1/p}=\alpha_{0}(h\nu -E_{g})$$

If *α_o_* is a constant, the direct transition is indicated by the exponent *p*’s value (*p* = 1/2). Then Tauc’s equation is depicted in Figure 10 as the plot of *(hν)* versus *(αhν)^2^*. The band gap energy $E_{g}^{opt}$ is determined at the intercept of the extrapolated line and *(αhν)^2^* = 0. The computed $E_{g}^{opt}$ values for Cu-doped and undoped crystalline ZnTe films are plotted in Figure 11.

Additionally, as demonstrated in Figure 11, the band gap energy reduced as more Cu was integrated into the ZnTe lattice. The recombination of an excited electron in the ZnTe lattice’s CB, which offered the best position for this recombination with opposing charge carriers, may be related to the defective broadband tailing generated by the formation of localized energy states close to the band boundaries. However, at a 10% Cu concentration, there was a minor increase in the band gap to 1.965 eV, which may be associated with the Burstein–Moss effect [43]. The Burstein–Moss shift is a phenomenon that increases a semiconductor’s perceived band gap and moves the absorption edge to higher energies when some states close to the conduction band are occupied. When the electron distribution is degenerate, as it is in some degenerate semiconductors, this effect is observed.

### 3.4. Electrical Characteristics

Hall effect measurements were used to determine the carrier type, carrier concentration and mobility of the ZnTe:Cu films with various Cu concentrations. As was observed, a positive Hall coefficient indicated that these films performed with p-type conduction. The data for the Hall mobility, carrier concentrations and the resistivity of undoped and Cu-doped ZnTe films with varied Cu incorporation levels at room temperature are shown in Figure 12. The results indicate that both the Hall mobility and carrier concentration rose as the Cu was increased from 0 at.% to 10 at.%, peaking at 1.85 × 10^18^ cm^−3^ and 54.21 cm^2^Vs^−1^, respectively, when the Cu was equal to 8 at.% (10 wt.%). The decrease in grain boundary scattering brought on by the rise in grain size was due to the increase in the Cu doping level, resulting in an increase in the carrier concentration and Hall mobility. The resistivity of the undoped and Cu-doped ZnTe thin films at room temperature was reduced as a result of the increased mobility, and reached the optimum minimum value at 0.08 (Ω.cm) when the Cu was equal to 8 at.% (10 wt.%). When the concentration of Cu^2+^ ions rise rather than the concentration of Zn^2+^ ions, extra electrons are forced to the conduction band, causing an additional decrease in resistivity. This effect clarifies why the resistivity decreased as the amount of Cu increased. Figure 13 plots the resistivity of the ZnTe:Cu films as a function of temperature. The figure shows that the resistivity vs. temperature curve of the ZnTe:Cu films was lowered with an increasing Cu incorporation level.

To comprehend the mechanism of conduction, Figure 14 analyzes and presents the temperature-supported dc conductivity data (in the manner of ln(σ) versus 1000/T descriptions) of the crystalline ZnTe:Cu films with varied Cu contents. There is no doubt that the conductivity has a non-linear temperature dependence. Additionally, it is observable that the dc electrical conductivity of the ZnTe:Cu films increases along with temperature. As a result, over the whole measurement temperature range of 300 K to 500 K, all the Cu-doped ZnTe films displayed semiconducting-like behavior. Moreover, the conductivity rose together with the level of Cu inclusion in the ZnTe film, because the larger grains caused less grain boundary scattering. The performance of the ZnTe layer in the p–n junction and the solar cell is predicted to be enhanced by raising the Cu inclusion level in the ZnTe film as the conductivity increases. CdSe:Cu films [44], CdTe:Cu films [45] and CdSe:Ag films [46] have all demonstrated a similar behavior.

The results reveal two unique zones with two slopes at low and high distinct temperature ranges, according to the formula *σ =σ_o_ exp(−E_a_/k_B_T)*,), where *k_B_* is the Boltzmann constant and *E_a_* is the thermal activation energy. Two conduction mechanisms are described here for the carrier transport in the ZnTe:Cu films. Thus, the slope of the curves in the two regions is used to calculate the high-temperature activation energy *E_a_*_1_ (380–500 K) and the low-temperature activation energy *E_a_*_2_ (300–379 K) of the ZnTe:Cu films, using the equation *σ =σ_1_ exp(−E_a1_/kT) + σ_2_ exp(−E_a2_/kT)*, where σ_1_ and σ_2_ are the pre-exponential factors.

The calculated values of *E*_a1_ and *E*_a2_ for the ZnTe:Cu films are plotted versus the Cu content in Figure 15. The activation energy at low temperatures (*E*_a2_) may be caused by low-temperature conductivity, due to carrier hopping between the localized states above the edge of the valence band and the extended states in the conduction band [44,45,46]. Normal band conduction in the extended states is the main mode of conduction in the high-temperature region.

### 3.5. ZnTe Layer and ZnTe:Cu Buffer Layer Effects on CdS/CdTe Solar Cell Performance

The diagram of the ITO/CdS/ZnTe/ZnTe:Cu/p-CdTe solar cell configuration is shown in Figure 16. Figure 17a,b shows how the ZnTe layer and ZnTe:Cu with a Cu concentration equal to 8% at optimal carrier concentration, mobility and lowest resistance effect the performance of the CdS/CdTe solar cells. An ITO/ZnTe/p-CdTe with the insertion of a CdS buffer layer between the ITO and i-ZnTe is the structure shown in Figure 17a, which exhibits strong rectifying diode characteristics. The PCE measured for the open circuit voltage (V_oc_), short circuit current density (J_sc_) and short circuit area was 0.77 V, 0.016 A/cm^2^ and 19.6%, respectively. This figure also demonstrates that after the ZnTe:Cu layer was inserted as a p-layer for the connections on the p-CdTe substrate, the V_oc_, J_sc_ and PCE of the ITO/CdS/ZnTe/ZnTe:Cu/p-CdTe increased to 0.815 V, 0.020 A/cm^2^ and 26.1%, respectively. According to Figure 17b, the maximum power for both solar cells was equal to 0.0071 and 0.0095 W for the ZnTe and ZnTe:Cu cells, respectively. We think that this change in the device’s performance may be caused by the high transmittance value and low reflection. Furthermore, compared to the Zn0.92Cu0.08Te-layered cells, the J_sc_ of the ZnTe-layered cells was lower. The development of photoelectric properties may be the cause of the raised V_oc_ and J_sc_ values for ZnTe, given that many photons were absorbed into the CdS/CdTe solar cell as a result of the decrease in reflectivity, which caused more absorption in the absorption layer. Better J_sc_ and V_oc_ values are produced by photo-generated carriers. Furthermore, the improvement in crystallinity within the Zn_0.92_Cu_0.08_Te layer, due to an increase in grain size and a decrease in resistivity, may be connected to the factors which produced the highest J_sc_ and V_oc_ values. The reduced resistivity of the Zn_0.92_Cu_0.08_Te layer may aid in the development of the filling factor (FF) of the CdS/CdTe solar cells. The aforementioned research suggested that the Zn_0.92_Cu_0.08_Te buffer layer is beneficial for improving the PEC of the CdS/CdTe solar cells. According to the above-mentioned study, the Zn_0.92_Cu_0.08_Te buffer layer is valuable for the resulting increase in the PE of the CdS/CdTe solar cells.

## 4. Conclusions

In conclusion, using an electron beam evaporation technique, pure and Cu-doped ZnTe films were evaporated onto glass substrates. The results of the XRD analysis showed that all of the films had a zinc-blende cubic nature. It was shown that the microstrain was reduced from 0.73 × 10^−3^ to 0.58 × 10^−3^ while the crystallite size increased from 18 nm to 33 nm when the Cu incorporation amount increased from 0 wt% to 10 wt%. Because the Cu was incorporated into the matrix of the ZnTe films, Auger Peaks were visible in the XPS studies. This demonstrated the existence of a Cu^2+^ oxidation state. The refractive index was computed using the envelope method, and it was found that the refractive index increased with an increase in the Cu-doping concentrations. When the Cu concentration increased from 0 at% to 8 at%, the optical band gap energy was observed to first decrease from 2.225 eV to 1.941 eV and then slightly increase to 1.965 eV, which may be associated with the Burstein–Moss effect. The observed boost in the dc electrical conductivity with a greater Cu doping was assumed to be the result of the larger grain size, which reduced scattering of the grain border. There were two carrier transport conduction processes visible in the structured ZnTe:Cu films. The conductivity of the ZnTe:Cu films was low-temperature dependent according to Mott’s adjustable range hopping conduction mechanism model. Use of the Hall effect experiments illustrated that all the films behaved in a p-type style. The outcomes further showed that as the Cu doping level rose, the carrier concentration and the Hall mobility also rose, and reached an optimal concentration of Cu at 8 at.%, because grain boundary scattering decreases with grain size. The impact of ZnTe and ZnTe:Cu (at Cu 8 at.%) layers on the performance of the CdS/CdTe cells was also examined. The ITO/ZnTe/p-CdTe structure’s open circuit voltage (V_oc_) and short circuit current density (J_sc_) were measured at 0.77 V and 0.016 A/cm^2^, respectively, and it had excellent rectifying diode characteristics. The V_oc_ and J_sc_ of the ITO/CdS/ZnTe/ZnTe:Cu/p-CdTe increased to 0.815 V and 0.020 A/cm^2^, respectively, with the introduction of a second ZnTe:Cu layer as a p-layer for the connections on the p-CdTe substrate.

## Figures and Tables

**Figure 1 materials-16-03082-f001:**
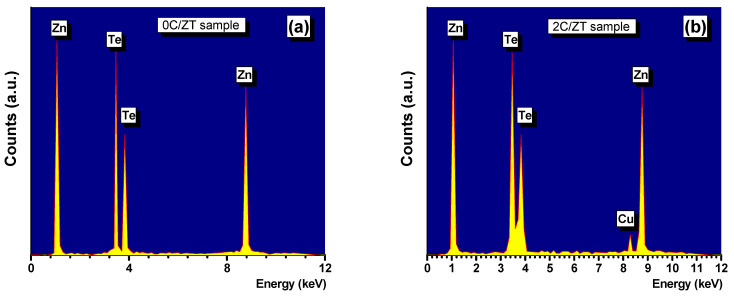
EDAX of (**a**) ZnTe and (**b**) Zn_0.98_Cu_0.02_Te thin films.

**Figure 2 materials-16-03082-f002:**
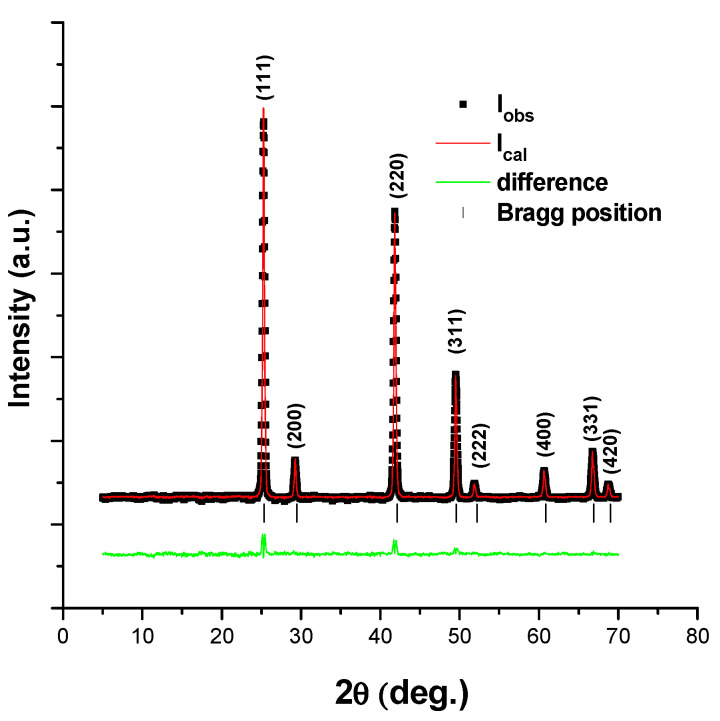
Rietveld refinement of ZnTe powder sample.

**Figure 3 materials-16-03082-f003:**
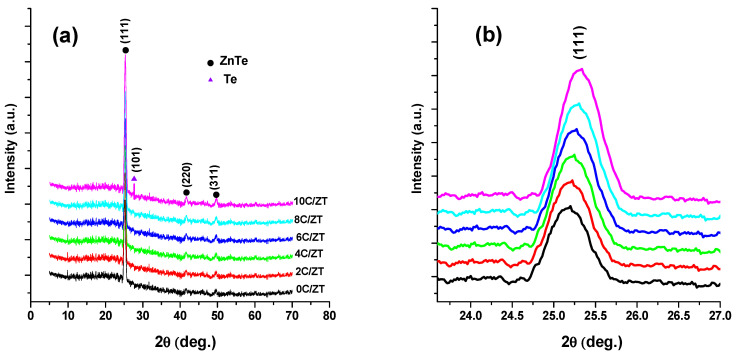
(**a**) XRD of ZnTe:Cu thin films, (**b**) amplification of XRD peaks with index (111).

**Figure 4 materials-16-03082-f004:**
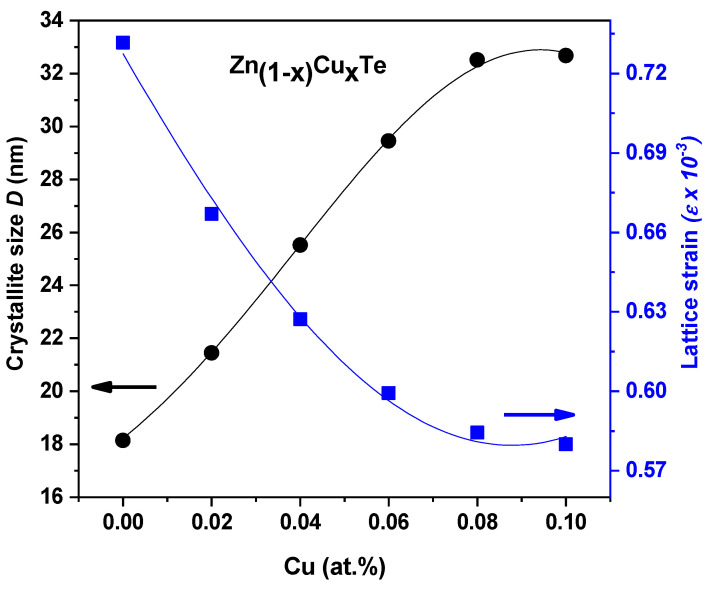
Crystallite size and lattice strain of the ZnTe:Cu thin films.

**Figure 5 materials-16-03082-f005:**
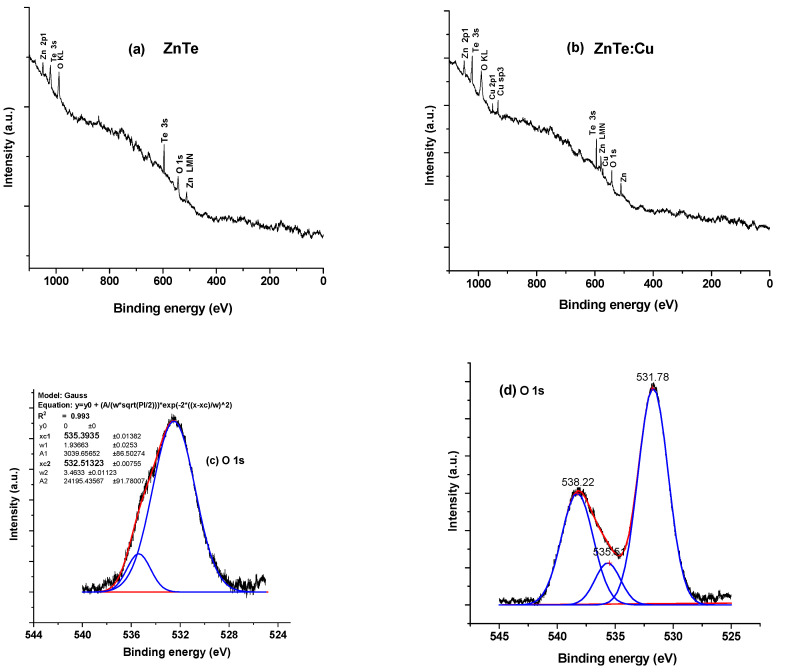
(**a**,**b**) XPS spectra of undoped ZnTe and doped with 10 at. % Cu thin films, (**c**,**d**) deconvoluted XPS analysis of Oxygen-1s for undoped and Cu-doped ZnTe thin-film samples, (**e**,**f**) deconvoluted XPS analysis of Zinc-2p for un-doped and Cu-doped ZnTe thin-film samples and (**g**,**h**) deconvoluted XPS analysis of Tellurium-3d for un-doped and Cu-doped ZnTe thin films.

**Figure 6 materials-16-03082-f006:**
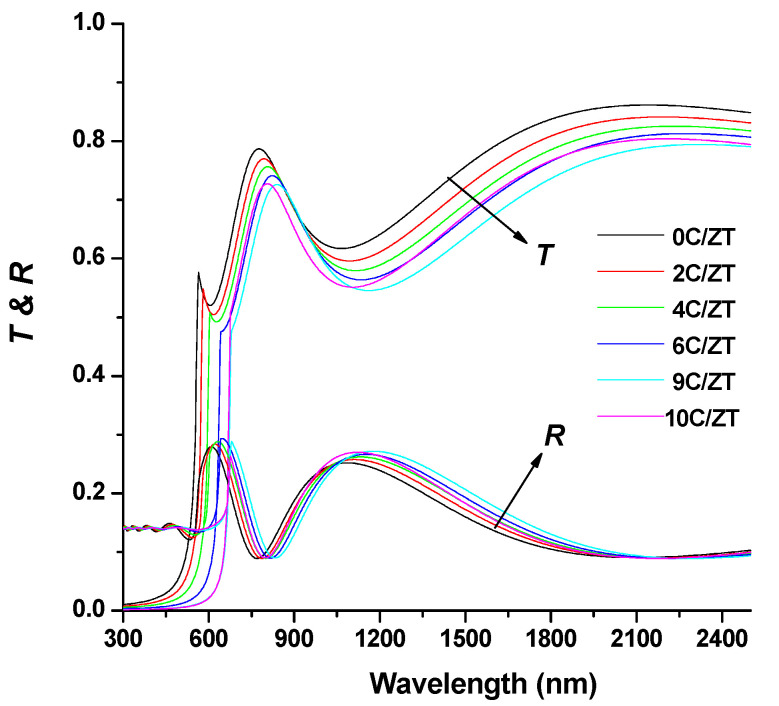
The spectral variation in the transmittance and reflectance of pure and Cu-doped ZnTe films with various Cu contents.

**Figure 7 materials-16-03082-f007:**
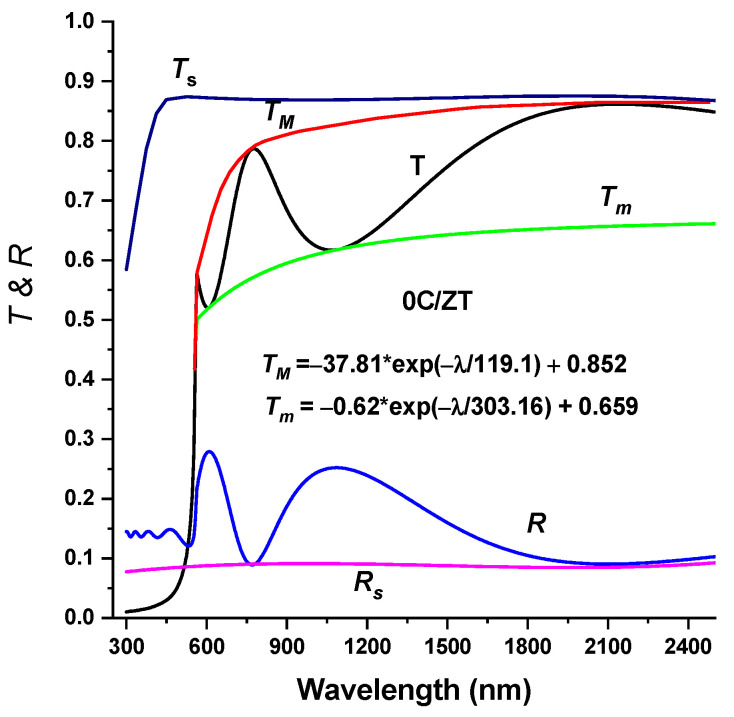
Optical transmission and reflectance spectrum of the ZnTe thin film. The top and bottom transmittance envelopes, *T*_M_ and *T*_m_ and *T*_s_ and *R*_s_, are both shown.

**Figure 8 materials-16-03082-f008:**
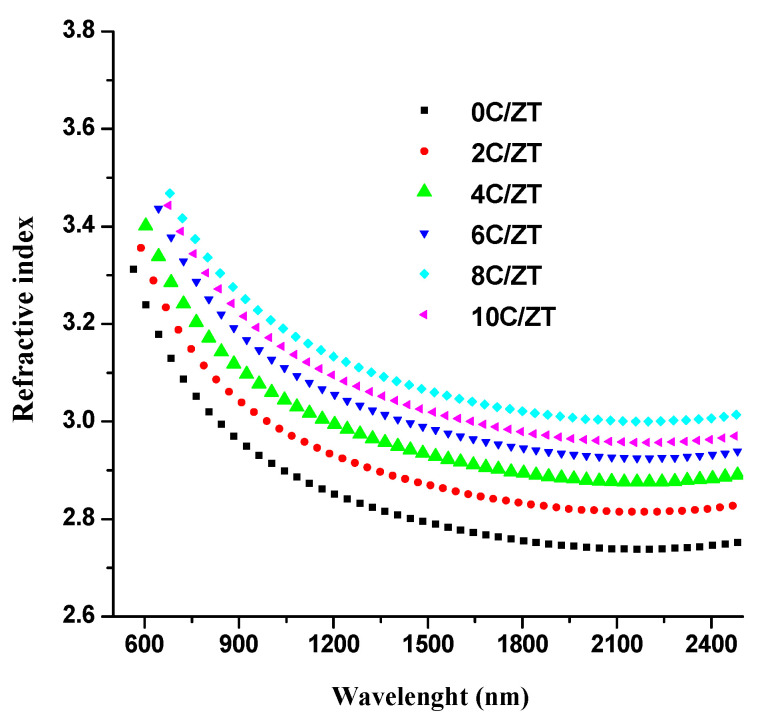
Dispersion of the refractive index of the ZnTe:Cu thin films obtained from Swanepoel’s method.

**Figure 9 materials-16-03082-f009:**
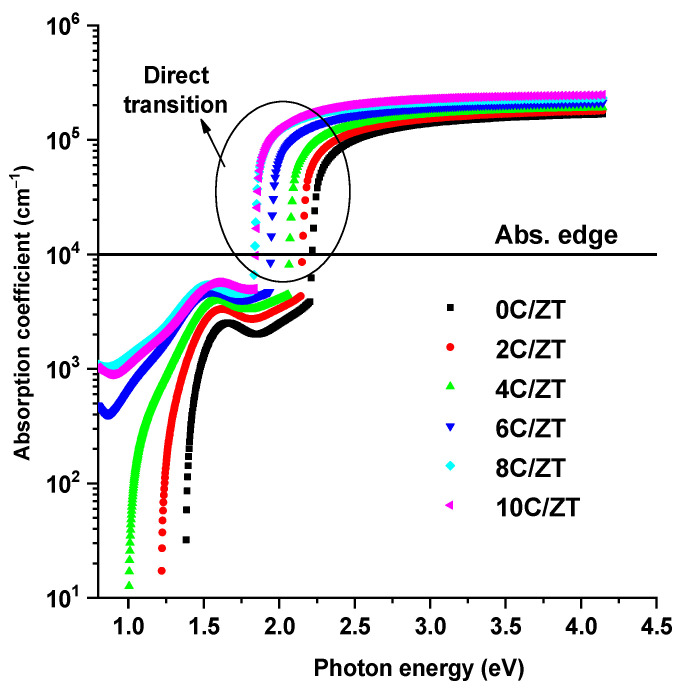
The absorption coefficient versus photon energy of the ZnTe:Cu films.

**Figure 10 materials-16-03082-f010:**
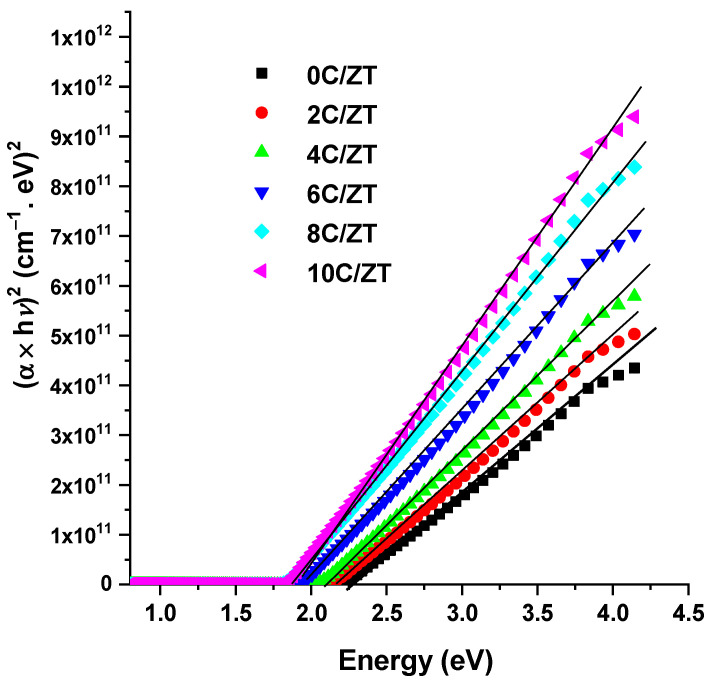
*(αhν)^2^* versus *hν* of the undoped and Cu-doped ZnTe films at different Cu contents.

**Figure 11 materials-16-03082-f011:**
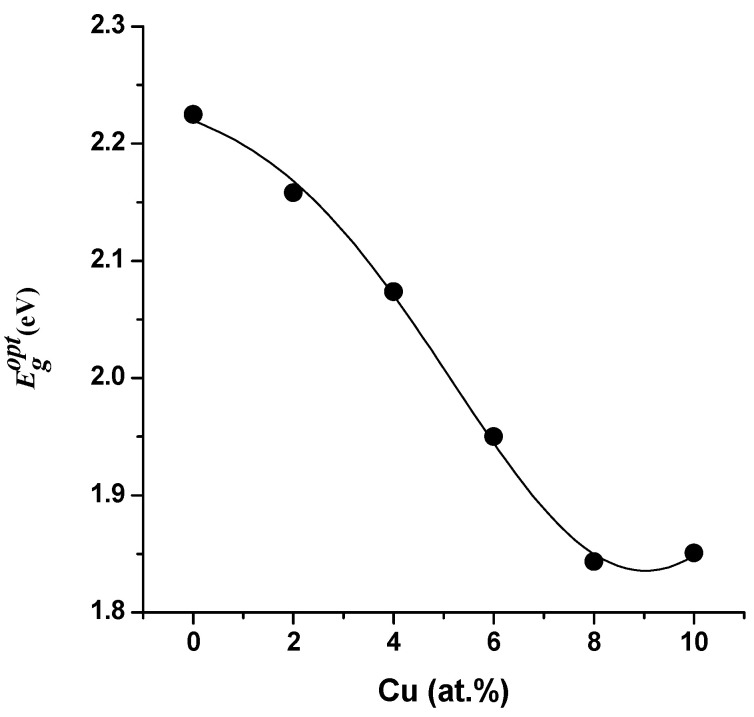
Energy gap as a function of the Cu concentration of the ZnTe:Cu films.

**Figure 12 materials-16-03082-f012:**
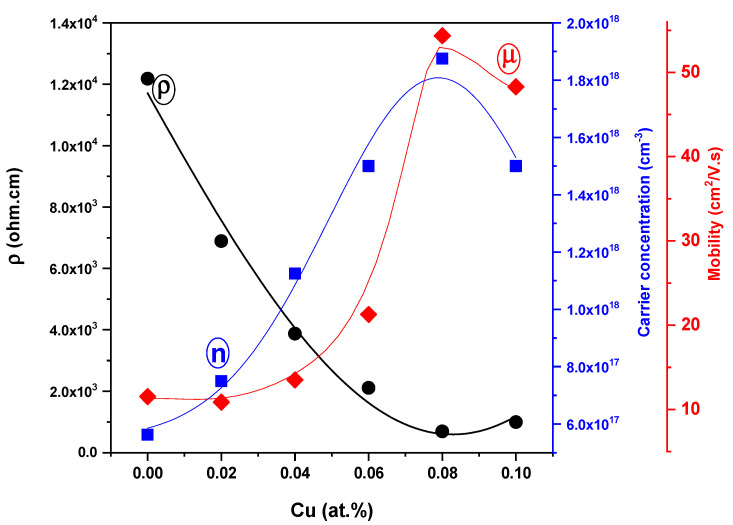
The resistivity ρ, carrier concentrations n, Hall mobility μ and RT of undoped and Cu-doped ZnTe film with various Cu incorporation levels.

**Figure 13 materials-16-03082-f013:**
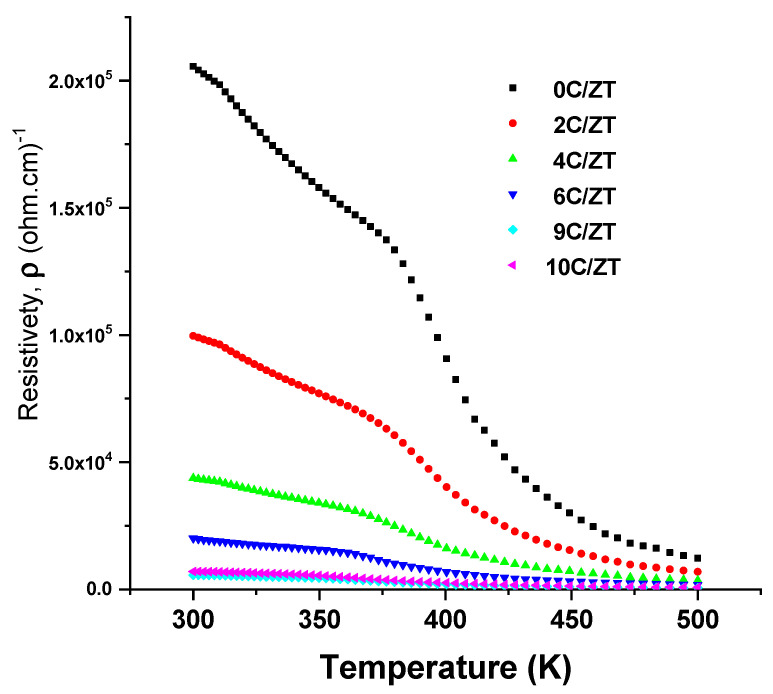
The resistivity ρ versus temperature of the undoped and Cu-doped ZnTe films.

**Figure 14 materials-16-03082-f014:**
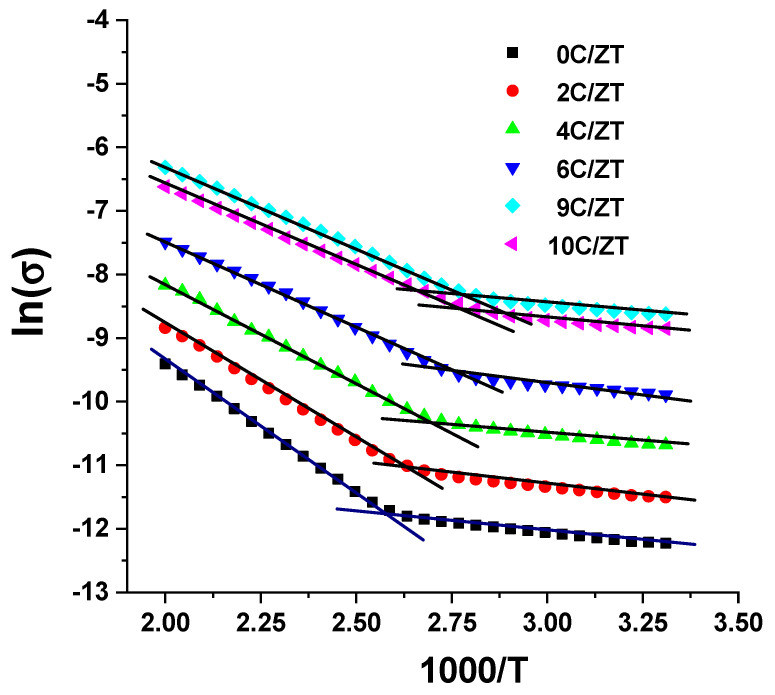
ln(σ) versus 1000/T of undoped and Cu-doped ZnTe films with various Cu incorporation levels.

**Figure 15 materials-16-03082-f015:**
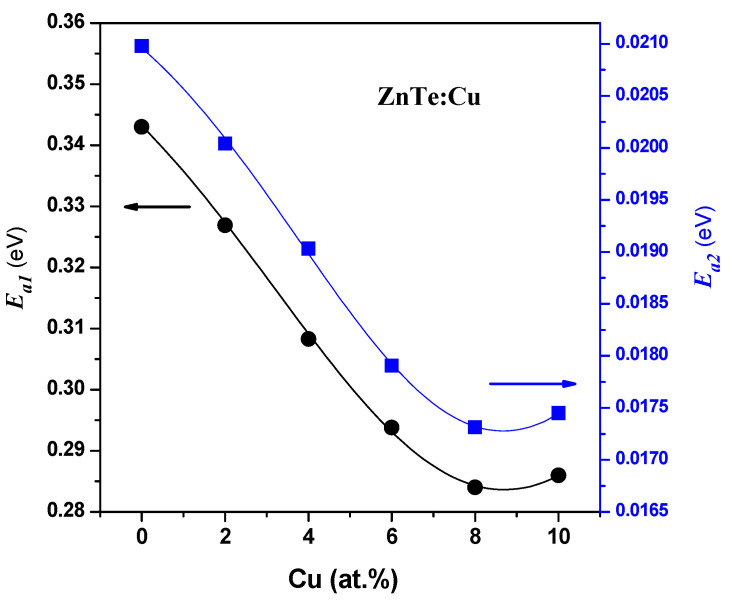
Dependence of activation energies on Cu content (at.%).

**Figure 16 materials-16-03082-f016:**
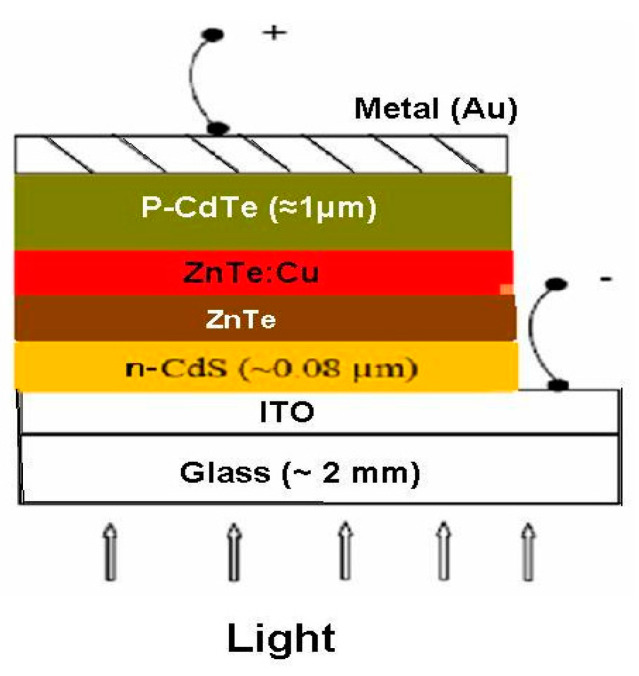
The basic structure of the glass/ITO/CdS/ZnTe/ZnTe:Cu/CdTe/metal thin-film solar cell diagram.

**Figure 17 materials-16-03082-f017:**
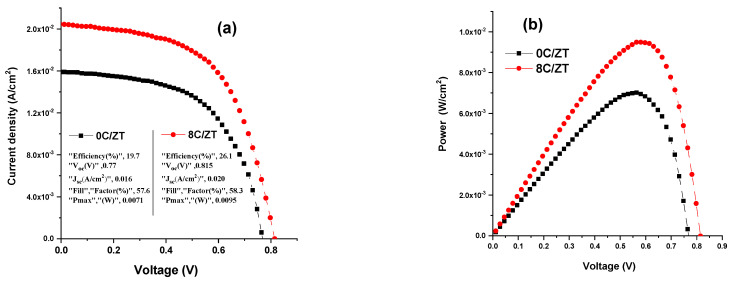
Characteristic curves of the solar cells’ (**a**) current density versus voltage and (**b**) power versus voltage for ITO/CdS/i-ZnTe /p-CdTe and ITO/CdS/i-ZnTe/ZnTe:Cu/p-CdTe.

**Table 1 materials-16-03082-t001:** Weight percent and atomic percent for the undoped ZnTe and ZnTe:Cu obtained using EDAX.

Sample’s Code	Weight %			Atomic %		
	Zn (%)	Te (%)	Cu (%)	Zn (%)	Te (%)	Cu (%)
0C/ZT	34.07	65.93	0	50	50	0
2C/ZT	32.29	65.11	2.59	48	50	2
4C/ZT	30.57	64.31	5.12	46	50	4
6C/ZT	28.88	63.52	7.59	44	50	6
8C/ZT	27.24	62.76	10	42	50	8
10C/ZT	25.63	62.01	12.35	40	50	10

**Table 2 materials-16-03082-t002:** Crystallographic parameters of ZnTe and the ZnTe:Cu thin films.

Sample’s Code	Diffraction Peak 2θ^o^	Interplanar Distance, d (Å)	Lattice Parameter, a (Å)	Crystallite Size (nm)	Lattice Strain (10^−3^)
0C/ZT	25.183	3.5321	6.1178	18.1	0.732
2C/ZT	25.202	3.5295	6.1133	21.4	0.667
4C/ZT	25.227	3.5261	6.1073	25.5	0.627
6C/ZT	25.255	3.5222	6.1007	29.5	0.599
8C/ZT	25.305	3.5154	6.0888	32.5	0.584
10C/ZT	25.318	3.5136	6.0857	32.7	0.58

## Data Availability

The data generated and/or analyzed during the current study are not publicly available for legal/ethical reasons but are available from the corresponding author on reasonable request.

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
