# Peer review of "Resulting Effect of the p-Type of ZnTe: Cu Thin Films of the Intermediate Layer in Heterojunction Solar Cells: Structural, Optical, and Electrical Characteristics"

_materials, 2023, doi:10.3390/ma16083082_

Round 1

Reviewer 1 Report

The review entitled “Resulting Effect of p-Type of ZnTe: Cu Thin Films of the Intermediate Layer in Heterojunction Solar Cells:      Structural, Optical, and Electrical Characteristics” focus on the increase of the Cu doping on ZnTe layers affected the efficiency of CdS/CdTe solar. This manuscript is written very poorly. It needs significant corrections and correlation with the results. But, the content of the paper is promising and suitable for publication in this journal. It Below, I have provided some minor remarks on the text before publication.

1.     Remove the type and grammar errors that exist in the context; please examine carefully.

As example,

Page 1 – “microstrian” = “microstrain”

Page 2 - “opto-electronic” = “optoelectronic”

Page 5- “interplaner spacing (dhkl)” = “interplanar spacing (dhkl)”

2.     Page 2-Using a traditional solid-state reaction method, various (ZnTe)1-x(Cu2Te)x compositions (with x = 0, 0.02, 0.04, 0.06, 0.08 and 0.10) were created.”. These sentences should be modified. You can say fabricated. It is not the proper way to say created.

3.     Page 2- “VI-II composites are important semiconductors for opto-electronic, photoconductive, and photoelectric devices and are frequently employed in solid-state electronics [1-3]. Because to its low electron affinity of 3.53 eV and ideal energy gap of 2.24 eV at ambient temperature, ZnTe has attracted particular interest for use in solar cells”. These sentences should be modified. It is not clear.

4.     Put one or two sentences describing why VI-II composites are more important semiconductors than lead halide-based perovskite semiconductors for photoconductive and photoelectric devices applications and cite related articles. As an example, cite these articles, J. Mater. Chem. A, 2020,8, 27-54, J. Mater. Chem. C, 2021,9, 15189-15200, J. Phys. Chem. C 2022, 126, 31, 13458–13466 etc.

5.     Page 5, “The combined effect of the host ZnTe crystal's lattice being enlarged by the replacement of Cu atoms with Zn atoms of the same ion size without affecting the crystal structure may be the cause of the observed rise in D and decrease in.” – this sentence not correct. Divide it into two sentences and explain it adequately. What do you mean by “cause of the observed rise in D and decrease in”?

6.     Resistivity unit on the Y axis in Figure 13 is not correct. Update the figure.

7.     Figure 17 is not clear. Texts are nonuniform. Please make a new figure and update it.

8.     There is many format mismatch and careless citation in the reference section. Use proper format and update it carefully. As an example, Ref no 16- 23 is in another format. After [25], there is ref no [29]. [26] to [33] again, bad formatting. 35, 37-48 ref citation format is horrible. 

Reviewer 2 Report

This manuscript can be published with minor corrections.

- Check line 167. 

-Include a cross section SEM for the solar cell.

Reviewer 3 Report

The paper of Moustafa Ahmed et.al is devoted to the material research (structural, optical and transport properties) of ZnTe:Cu layers, which can be of interest for photovoltaic applications. The best type of ZnTe:Cu layers (concentration of Cu 8%) was determined and photovoltaic properties of “ZnTe:Cu/CdTe/CdS” solar cells were measured.

Reviewer has the following remarks:

1) It is not clear from the text of the paper what could be the advantage of proposed II-VI solar cells over conventional silicon based solar cells. Measured value of photovoltaic efficiency (20%) is less than that in silicon solar cells. It is worth noting that silicon solar cells are much cheaper and easier in production. The authors must clarify the prospects of their structures in photovoltaic applications.

2) Two tables (Table1 and Table2) are mentioned in the text of the paper, but the tables themselves were not found in the paper. It is necessary to add these tables to the paper.

3) EDAX spectra in Figure 1 look suspiciously identical (even the noise looks identical!), except for the Cu-peak. Provide real spectra for these samples or explain the noise identity between peaks.

4) XRD spectra in Figure 3b look suspiciously identical (even the noise looks identical!), except for the (111)-peak. Provide real spectra for these samples or explain the noise identity between peaks.

5) In section 3.1 (XRD), a conclusion is given about the grains enlargement. This conclusion must be confirmed also by microscopy direct methods.

6) The text contains a large amount of typos, for example: Page 5, Line175: “decrease in.” –> “decrease in strain.”; Page 5, Line167: “where ?=??? is”

7) The English should be improved in the paper.

Round 2

Reviewer 3 Report

Authors should pay attention to following the rules of the journal format. In some places in the paper different fonts are used. In Figure 5, the arrangement of individual fragments is broken.